# Mutant IDH in Gliomas: Role in Cancer and Treatment Options

**DOI:** 10.3390/cancers15112883

**Published:** 2023-05-23

**Authors:** Georgios Solomou, Alina Finch, Asim Asghar, Chiara Bardella

**Affiliations:** 1Institute of Cancer and Genomic Sciences, College of Medical and Dental Sciences, University of Birmingham, Birmingham, B15 2TT, UK; 2Division of Academic Neurosurgery, Department of Clinical Neurosciences, University of Cambridge, Cambridge CB2 0QQ, UK; 3Wellcome MRC Cambridge Stem Cell Institute, University of Cambridge, Cambridge CB2 0AW, UK

**Keywords:** isocitrate dehydrogenase, cancer metabolism, hydroxyglutarate, gliomas, oncometabolites

## Abstract

**Simple Summary:**

Isocitrate dehydrogenase (*IDH*) is the most mutated metabolic gene in human cancer. Mutations in this gene have been identified in a high percentage of gliomas, in acute myeloid leukemia (AML) and in many other malignancies. Mutant *IDH* causes tumour initiation, possibly through accumulation of the oncometabolite D2-hydroxyglutarate, a product of normal metabolism with limited known function in mammals. Many studies have tried to elucidate the biological consequences of mutant IDH and to find effective inhibitors targeting this enzyme. This review delves into the cellular, biochemical and molecular consequences of mutant IDH and the therapeutic strategies for treating IDH mutant cancers.

**Abstract:**

Altered metabolism is a common feature of many cancers and, in some cases, is a consequence of mutation in metabolic genes, such as the ones involved in the TCA cycle. Isocitrate dehydrogenase (*IDH*) is mutated in many gliomas and other cancers. Physiologically, IDH converts isocitrate to α-ketoglutarate (α-KG), but when mutated, IDH reduces α-KG to D2-hydroxyglutarate (D2-HG). D2-HG accumulates at elevated levels in IDH mutant tumours, and in the last decade, a massive effort has been made to develop small inhibitors targeting mutant IDH. In this review, we summarise the current knowledge about the cellular and molecular consequences of IDH mutations and the therapeutic approaches developed to target IDH mutant tumours, focusing on gliomas.

## 1. Introduction

The isocitrate dehydrogenase (IDH) family includes three enzymes, IDH1, IDH2 and IDH3, ubiquitously expressed in human cells. Although all these enzymes catalyse the same chemical reaction, i.e., the oxidative decarboxylation of isocitrate (ICT) to α-ketoglutarate (α-KG), each enzyme has unique biological properties. IDH1 is localised to the cytoplasm and peroxisomes of the cell, where it has a role in glucose and lipid metabolisms and protects against reactive oxygen species (ROS) [1,2,3,4]. IDH2 and IDH3 are both found in the mitochondrial matrix [5]; here, IDH3 functions as a central enzyme of the tricarboxylic acid cycle (TCA), whereas IDH2 exerts its regulatory function on the TCA cycle and protects against oxidative stress [6]. IDH1 and IDH2 are homodimers; IDH3 is instead a heterotetrameric protein composed of two catalytic α subunits required for ICT binding, a β subunit, which has a structural role in facilitating the enzyme subunits assembly, and the γ subunit, which, through binding of citrate and ADP, functions as an allosteric modulator [7,8]. The chemical reaction catalysed by IDH1 and IDH2 is reversible and requires the binding of nicotinamide adenine dinucleotide phosphate (NADP^+^) as a co-factor, which is converted into NADPH [9], while the chemical reaction of IDH3 is irreversible and involves the binding of NAD^+^, which is reduced to NADH. Moreover, unlike IDH1 and 2, the chemical reaction catalysed by IDH3 is tightly regulated by substrate availability and allosteric effectors (citrate, ICT, adenosine 5′-diphosphate (ADP), nicotinamide adenine dinucleotide (NAD^+^), Mg^2+^/Mn^2+^and Ca^2+^), product inhibition (NADH and α-KG) and competitive feedback inhibition (adenosine 5′-triphosphate (ATP)) [10] to avoid unnecessary usage of ICT and increase in α-KG.

In recent decades, recurrent somatic mutations in *IDH1* and *IDH2* have been identified in many cancers, such as low-grade gliomas (LGG) (>70%), secondary glioblastoma (diffuse astrocytoma, WHO CNS grade 4, according to the 2021 guidelines) (55–88%), primary glioblastoma (5–14%) [11,12,13,14,15,16,17], cartilaginous and bone tumours (20–80%) [18,19,20,21,22,23,24], intrahepatic cholangiocarcinoma (6–30%) [25,26,27,28,29,30,31,32,33], acute myeloid leukaemia (15–30%) [34,35,36,37,38,39,40,41,42], as well as, to a lower extent, myelodysplastic syndrome and angioimmunoblastic T-cell lymphoma [43,44,45,46], and other solid tumours [47,48,49,50,51,52,53,54,55,56,57,58,59].

In contrast to *IDH1* and *IDH2*, no tumour-associated mutations for any *IDH3* genes (*IDH3A*, *IDH3B* and *IDH3G*) have been reported [60].

*IDH* changes are missense mutations, which predominantly affect the aminoacidic residues R132 and R100 in IDH1, and the equivalent residues R172 and R140 in IDH2 [11,12,34,47]. These arginine residues are all localised in the substrate binding site of the enzyme, where they form hydrophilic interactions with the α/β-carboxylate of the isocitrate [7,35,61]. Specifically, in the mutant IDH1, the highly positively charged residue arginine 132 is replaced by lower polarity amino acids, such as histidine, cysteine, glycine or lysine. When the protein is mutated, there is a switch between the open and closed structure of the enzyme [35,61], resulting in a weak affinity for isocitrate and an increased affinity for α-KG and NADPH, which are preferentially bounded [35,61]. Since IDH mutations are monoallelic changes, they result in an enzyme composed mainly of wild-type and mutant monomers (Figure 1). Thus, in IDH mutant cells, the wild-type part of the dimer leads to the conversion of isocitrate to α-KG, producing NADPH, whereas the mutant monomer exhibits neomorphic activity, as it converts α-KG to the D2-enantiomer of hydroxyglutarate (Figure 1) (D2-HG; the L2-enantiomer, L2-HG, is not significantly produced), using NADPH as a co-factor [61]. Consequently, in IDH mutant tumours, D2-HG accumulates to millimolar concentrations, while, due to the enzymatic loss of function [12], the α-KG decreases (Figure 1) [35,38,61,62]. In addition to its role as a biomarker in IDH mutant malignancies, D2-HG is thought to be an oncometabolite [63], mainly for its role in competitively inhibiting the enzymatic activities of α-KG-dependent dioxygenases, such as histone lysine demethylases (KDMs), or ten-eleven translocation methylcytosine (TET) family enzymes (Figure 2) [63,64]. Consequently, IDH mutations result in the hypermethylation of DNA and histones, which is considered one of the leading mechanisms of tumourigenesis. Indeed, also the reduction of α-KG caused by mutant IDH [65,66,67,68,69,70,71], might contribute to its tumorigenic effect [63].

*IDH1* and *IDH2* mutations appear mutually exclusive [36,40], with some rare exceptions [13,36,42]. Interestingly, the nature of the various IDH substitutions differs among the cancer types, i.e., IDH1 is more frequently mutated in solid tumours, while IDH2 is mainly mutated in some blood cancers. Moreover, specific IDH variants seem more frequent in some tumour types. For instance, in LGG, most IDH mutations occur in the *IDH1* gene, which is mainly converted to the R132H variant. *IDH1* is most frequently mutated in chondrosarcoma and cholangiocarcinoma, specifically in the variant R132C [72]. Interestingly, patients harbouring *IDH1^R132H^* mutated tumours have lower levels of genome-wide DNA methylation, an increased gene expression and the worst prognosis compared to patients affected by tumours with other IDH1/2 mutations [72].

The reason for the tissue specificity of the various IDH mutations in different cancers is unclear. However, specific IDH mutations differ in their capacity to produce D2-HG [73,74]. For instance, IDH1^R132H^ has a relatively low ability to produce D2-HG compared to other mutations, such as IDH1^R132C^, which is highly more efficient [73,74]. Potentially, each tumour type may require a particular amount of D2-HG to initiate tumour formation, as diverse levels of D2-HG might be required to impact tissue-specific enzymes/pathways driving cancer initiation and progression. For instance, not all α-KG-dependent dioxygenases are equally well inhibited by the D2-HG, and some of them, such as TET2 enzymes, require high levels of D2-HG to be inhibited [63].

It is well recognised that IDH mutations have a biological impact both intracellularly and as part of the tumour microenvironment, favouring tumour formation and recurrence. In recent decades, many functional studies have focused on understanding the molecular mechanisms responsible for the formation of IDH mutant tumour. These studies mainly focus on the functional significance of D2-HG accumulation, but despite many efforts, the biological mechanism driving tumour formation is not completely clear. Considering that current therapies fail to demonstrate improved outcomes, IDH-induced biochemical alterations should be adequately understood and assessed as potential targets. Here, we summarise the current knowledge of IDH-mutated human malignancies, focusing on glioma, discussing the oncogenic role of mutant IDH in tumour formation and the therapeutic opportunities.

## 2. The Oncogenic Role of Mutant IDH in Tumour Formation and Progression

### 2.1. Metabolic Alterations

IDH mutations result in significant alterations and reprogramming of the cellular metabolic pathways. Indeed, the production of D2-HG leads to significant reduction and drainage of TCA cycle substrates, and thus, carbohydrate sources [61,65]. Consequently, the TCA cycle is forced to adjust and drain other carbohydrate sources to yield ATP [75,76,77].

In IDH mutant cells, a considerable reduction in α-KG has been reported by most, but not all studies [61,65,66,67,68,69,70,71,76]. Although mutant IDH consumes α-KG, cellular α-KG stocks can be replaced from other sources. For instance, glutamine derivatives can serve as key substrates to the TCA cycle to replenish the exhaustion of isocitrate and α-KG metabolites. Glutamate dehydrogenase 2, the enzyme responsible for converting glutamate to α-KG, has been found highly expressed in IDH mutant cells to possibly compensate for the reduction in α-KG consequent to IDH mutation [78]. In addition, IDH-mutated glioma cells are susceptible to the inhibition of glutaminase. This enzyme contributes to the lysis of glutamate, further adding to the notion that IDH-mutated cells are glutamate-dependent [79]. Additionally, the depletion of NADPH caused by the formation of D2-HG leads to the reduction in intracellular lipogenesis, resulting in significant dependence on exogenous lipid sources [2].

Compared to other tumours, IDH mutant gliomas show a distinctive metabolic behaviour. In most cancers, the need for an adequate energy supply is mediated by the increased expression of lactate dehydrogenase (LDH) [80]. LDH catalyses the transformation of pyruvate, the end product of glycolysis, to L-lactate [81]. L-lactate can serve as the primary fuel for the increased demands for energy to match the rapid proliferative potential of the cancer cells. On the contrary, in IDH-mutated gliomas, the LDH is silenced [82,83]. Hypermethylation of the promoter region of the *LDH* gene was the main reason for the lack of LDH expression [83,84]. 

Moreover, IDH mutant gliomas rely less on the glycolytic pathway and more on oxidative phosphorylation to produce energy than wild-type (WT) IDH1 gliomas [85]. The reliance of IDH mutant cells on oxidative phosphorylation might provide metabolic targets for future IDH mutant glioma therapies.

### 2.2. Redox Imbalance

IDH1 and IDH2 are essential sources of NADPH in the cytoplasm and mitochondria, respectively [86]. Mutations in *IDH* determine an increased affinity of the enzyme for NADPH and α-KG [35,61], leading to high consumption of NADPH and a consequent reduction in the cellular NADPH/NADP^+^ ratio. This disrupts the reducing equivalents of biochemical reactions in the cell, accumulating reactive oxygen species (ROS) [87,88]. Excessive ROS damage DNA, RNA, lipid and protein in the cell, disrupting enzymatic reactions and gene expression. ROS are involved in genomic instability, cellular motility and acquisition of invasive characteristics [89].

Consequently, ROS accumulation is fundamental and a hallmark of cancer biology, especially for IDH-mutated gliomas. Cells derived from IDH1 mutated gliomas exhibit strong oxidative stress, evident by the increased expression of manganese superoxide dismutase [90]. The elevated pressure is also confirmed by the fact that IDHmutated cells are prone to oxidative damage [91,92]. In the face of the raised oxidative profile, enhancing anti-oxidant pathways, such as the synthesis of glutathione, may be a valuable strategy to downplay the oncogenic effects of ROS [93].

### 2.3. Epigenetic Modifications

Epigenetic alterations affect gene expression without causing permanent changes to the DNA sequence as mutations would. Epigenetic changes range from DNA methylation, non-coding RNA molecules (miRNA, siRNA, piRNA and lncRNA) and covalent modifications to histones, such as acetylation, methylation and phosphorylation [94,95]. IDH mutations are very tightly linked with changes to the epigenetic landscape in cells, as their abnormal production of D2-HG competitively inhibits a range of enzymes involved in maintaining “normal” DNA methylation and histone modifications. This disruption leads to genome-wide epigenetic changes called hypermethylation and widespread gene expression alterations [96]. Understanding the pathogenesis of IDH mutant gliomas concerning the hypermethylation patterns might lead to identifying rational therapeutic targets.

Analysis of DNA methylation changes of glioblastoma (GBM) samples from *The Cancer Genome Atlas* (*TCGA*) database found a strict correlation between IDH mutation and the development of a unique DNA methylation signature [97,98]. This became known as the glioma cytosine-phosphate-guanine (CpG) island methylator phenotype (G-CIMP). The G-CIMP phenotype is considered a favourable marker of prognosis only in patients affected by IDH mutant gliomas [99]. However, increased hypermethylation on CpG genome sites has been partially found in other IDH mutant tumours compared to their wild-type countertypes [96]. A study by Unruh and colleagues demonstrated that the extent and targets of the DNA hypermethylation induced by mutant IDH are highly variable between various IDH mutant cancers (AML, cholangiocarcinomas, melanomas, gliomas), depending on the cellular contexts. This might explain why mutant IDH is a favourable prognostic marker specifically in gliomas and not in other IDH mutant tumours [96].

Methyltransferases and demethylases control the methylation pattern of the DNA. During the DNA demethylation process, the conversion of 5-methylcytosine (5-mC) to 5-hydroxymethylcytosine (5-hmC) is catalysed by the methylcytosine dioxygenases TET (TET1, TET2 and TET3 enzymes) in an α-KG-, oxygen- and iron-dependent manner. In addition, TETs concomitantly catalyse cytosine demethylation steps by converting 5-hmC to 5-formyl cytosine (5-fC) and then to 5-carboxyl cytosine (5-caC) (Figure 3). 5ca-C or 5-fC bases are subsequently recognised and excised by the enzyme thymine DNA glycosylase (TDG) and substituted by an unmodified cytosine (C) by base excision repair (BER) enzymes (Figure 3) [100]. Since TETs are α-KG-dependent dioxygenases, in IDH mutant gliomas their activity is inhibited by the D2-HG, which has a structural similarity to α-KG [63,101]. Therefore, in mutant IDH gliomas, the DNA demethylation process cannot occur, and a hypermethylated phenotype is present (Figure 2) [102,103]. Follow-up studies have shown that once hypermethylation occurs, it is irreversible, and thus, it plays a pivotal role in malignant transformation and recurrence [104].

Additionally, D2-HG aids histone methylation by inhibiting histone demethylases (Figure 2). One notable example are lysine-specific demethylases KDMs [63,64]. Histone methylation is predominantly regulated via histone methyltransferases, which add the methyl group, and demethylases (such as KDMs), which remove the methyl group. Because α-KG plays a catalytic role in these reactions, in an analogous way, as with the TETs mentioned above, the increased levels of D2-HG competitively block these reactions [40,63].

It is now recognised that histone and CpG island hypermethylation patterns are predominantly found in IDH mutant glioma stem cells (GSCs) [40]. Studies have shown that CpG hypermethylation leads to tumour suppressor genes (TSG) inactivation and altered gene expression related to cell differentiation (Figure 2) [36]. Thus, IDH mutations ultimately block cell differentiation and cell cycle regulation, leading to uncontrolled proliferation, which will cause the accumulation of new somatic mutations acquired over time. However, the development of glioma requires not only seeds (GSCs) with uncontrolled proliferation but also fertile soil (tumour microenvironment).

### 2.4. Tumour Microenvironment

The D2-HG produced by mutant IDH accumulates tomM ranges intracellularly [61,62] and extracellularly [105], causing a biochemical cascade of events inside the cell and important metabolic alterations in the surrounding cellular microenvironment. These metabolic changes may affect the behaviour of the non-neoplastic cells in the tissue in many ways to promote tumour growth and progression through non-cell intrinsic mechanisms.

#### 2.4.1. IDH Mutation and Tumour-Specific Immune Cells

The microenvironment surrounding glioma is rife with a variety of immune cells, including tumour infiltrating lymphocytes (TILs), natural killer (NK) cells, tumour-associated macrophages (TAMs), microglia, myeloid-derived suppressor cells (MDSCs), neutrophils, dendritic cells and fibroblasts [106].

Increasing evidence denotes that mutant IDH can decrease the levels of tumour immune surveillance, enabling cancer cells to escape it. Various glioma studies have shown that mutant IDH affect tumour immune-cell infiltration and function by modulating chemokine secretion. For instance, IDH1 mutation has been shown to reduce leukocyte chemotaxis into gliomas in a syngenic mouse model by repressing the expression of key chemoattractant cytokine genes, including *Cxcl-2*, *Ccl-2* and *C5* [107]. Other studies demonstrated that IDH mutant human gliomas had reduced levels of CD8^+^T lymphocytes compared to their WT counterpart; this was also confirmed using a syngenic mouse model of glioma, where expression of mutant IDH or treatment with 2-HG reduced the infiltration of cytotoxic CD8^+^T lymphocytes and levels of T-cell-associated chemokines within tumours. Interestingly, these effects were reversible using a specific inhibitor of mutant IDH1 [108]. Accordingly, other studies reported that glioma stem cells and astrocytes expressing IDH mutations could escape immune surveillance by causing epigenetic repression of natural killer cells’ ligand genes [109]. The inhibition of the immune response induced by mutant IDH might explain, at least in part, the difference in the aggressiveness of IDH mutant gliomas compared to the wild-type samples.

Many recent studies have also indicated that IDH mutation modifies TAM phenotypes to influence glioma growth. TAMs are the primary innate immune effector cells in malignant gliomas and can be in vivo separated into M_1_ with a pro-inflammatory signature and M_2_ with an anti-inflammatory signature [110]. Mutant IDH has been found to reduce the infiltration of TAMs in both IDH1 mutant mouse models and human glioma tissues [107]. Another study found that although the total number of TAMs is lower in IDH1 mutant compared to the WT IDH GBM samples, the remaining TAMs are more pro-inflammatory [111]. Conversely, another recent study reported that TAMs from patients with IDH mutant gliomas exhibit a more immunosuppressive phenotype than IDH wild-type samples [112].

Mutation in IDH1 plays an essential role in TAM activation. D2-HG can be taken up by TAMs using membrane transporters, favouring the activation of the M1 state of TAMs [113]. An in vitro assay combining human TAMs with glioma cells for 24 h showed that the presence of IDH1 mutation within the glioma cells caused an increase in the expression of M1 genes and decreased that of the M2 genes [113]. Moreover, in vitro and in vivo studies have shown that mutant IDH1 promotes TAM migration: the conditioned medium derived from IDH1 mutant glioma cells significantly increases TAM infiltration compared with IDH1 wild-type glioma cells. Accordingly, in vivo data showed that IDH1 mutant glioma cells have increased recruitment of TAM; these data support the anti-tumour functions of TAMs in vivo [113].

Other recent studies have shown that migration of both CD4^+^ and CD8^+^ T cells was inhibited by treatment with D2-HG, while T-cell proliferation and IFNγ production were both inhibited in a dose-dependent manner [114]. This provides direct evidence that D2-HG produced by mutant IDH is vital to immune suppression.

The role of D2-HG in immune suppression was further elucidated using genetically engineered CAR-T cells. CAR-T cells were generated, which were either a knock-out (KO) or overexpression (OE) of the D2-HGDH enzyme—D2-HG dehydrogenase—the enzyme responsible for the conversion of D2-HG to α-ketoglutarate. The D2-HGDH-KO and -OE CAR-T cells were tested using the cell line NALM6 overexpressing mutant IDH1. The results show that D2-HGDH-OE CAR-T cells had a profound rate of tumour cell death compared to D2-HGDH-KO CAR-T cells. Mice bearing mutant IDH1 tumours showed a significant increase in their survival following treatment with D2-HGDH-OE CAR-T cells. It was later revealed that the mechanism behind this was the reduction in D2-HG, and therefore, the removal of immune suppression, which increased memory cell differentiation and cytokine production [115,116].

A recent study provides new evidence for mechanistic insight into the effects of D2-HG on CD8^+^ T-cell inhibition [117]. Notarangelo et al. showed that D2-HG acts directly on CD8+ T cells to impair cytokine production and proliferation in a reversible and dose-dependent manner, suppressing this immune cell group. The study went further to show that D2-HG inhibits the ability of CD8^+^ T cells to perform glycolysis by direct inhibition of lactate dehydrogenase (LDH). In vitro studies using purified, recombinant LDH protein and exogenous D2-HG showed a non-competitive inhibition of the enzyme, leading to altered glycolytic flux in the CD8+ T cells. Application of D2-HG to CD8^+^ T cells led to a reduction in the production of IFN-γ and therefore also in its autocrine effects on gene expression. Encouragingly, the results in vitro are also found in vivo when analysing samples from IDH mutant tumours. These glioma samples demonstrate reduced IFN-γ-related gene expression and reduced lactate levels due to the inhibition by D2-HG [117].

Together, these data show that mutant IDH enzyme, via the aberrant production of D2-HG, leads to suppression of the immune system in terms of activation and infiltration. This leads to an immunologically “cold” tumour microenvironment, which improves overall patient survival but limits treatment options, as immune-targeted therapies are rendered ineffective. Further understanding of the difference in the immune system’s role in IDH WT and mutant gliomas will be essential to finding ways to target these tumours effectively.

#### 2.4.2. IDH Mutation, Intra-Tumour Hypoxia and Angiogenesis

Contrasting data exist in the literature regarding the ability of mutant IDH to increase the levels of hypoxia and the formation of new blood vessels within the tumour. It was initially hypothesised that reduced levels of α-KG [118] and the D2-HG produced by mutant IDH [63,64], may inhibit the prolyl hydroxylase (HIF-PHD) responsible for the ubiquitination and proteasomal degradation of hypoxia-inducible factor (HIF). Consequently, HIF could become stable in normoxic conditions and thus activate the expression of HIF-target genes, such as vascular endothelial growth factor (VEGF) [63,64,118]. Accordingly, HIF1α was upregulated in cells overexpressing mutant *IDH1*, in cells treated with exogenous 2-HG, in IDH mutant tumours and in brains of mouse embryos expressing *idh1* R132H [63,64,70,118]. However, D2-HG is only a relatively weak inhibitor of HIF-PHD [64], and following studies found no correlation between IDH mutation status and the expression of HIF1α in patients with glioma [119,120] or in the brains of adult mice where mutant *idh1* was expressed [71]. Furthermore, other studies demonstrated that D2-HG could stimulate rather than inhibit the activity of HIF-PHD. In some cellular contexts, such as human astrocyte, colorectal and erythroleukemia cell lines, the expression of IDH1 R132H reduced HIF levels [121,122]. Recently, other studies found decreased levels of VEGF in IDH mutants compared to WT GBM samples and that the expression of HIF2α and micro-vessel density were independent of IDH mutation status [123].

Similarly, a gene set variation analysis of a cohort of TCGA patients has shown that IDH1/2 mutations were associated with inhibiting angiogenesis signalling pathways. The same study also found that IDH mutant gliomas tend to have a lower relative cerebral blood volume compared to the IDH WT counterpart, indicative of a decreased angiogenesis of IDH mutant samples compared to the WT [124]. However, other findings reported that in primary GBM, the intra-tumoural levels of HIF1α and peripheral serum levels of VEGF were significantly higher in IDH1 mutated versus IDH WT samples [125]. Thus, these discrepancies may be due to experiments being conducted in cells originating from different tissues and organs but might also arise from the subset of genetic changes characterising the different types of tumours analysed. Consequently, a link between IDH mutation status and hypoxia-induced angiogenesis is not a straightforward phenomenon. The cellular context differences and the genetic changes between different tumours may cause contrasting experimental results. However, albeit controversial, these findings are interesting, as they highlight that although traditionally viewed as an oncogene, HIF may also act as a tumour suppressor under certain conditions.

#### 2.4.3. IDH Mutation and Tumour-Associated Epilepsy

Patients affected by IDH mutant gliomas present more frequently with epileptic activity than patients with IDH wild-type gliomas [126,127]. Unfortunately, the seizures shown by IDH mutant patients are frequently resistant to first-line epilepsy treatment [128]. Nevertheless, a patient affected by IDH1 mutant oligodendroglioma recently showed reduced tumour-associated epilepsy after treatment with the IDH1 mutant inhibitor AG-120 (ivosidenib) [129]. To explain the higher incidence of seizures in IDH mutant patients compared to the IDH wild-type patients, the neuroexcitatory properties of the D2-HG need to be considered. D2-HG is released by IDH mutant glioma cells into the tissue microenvironment, and it has structural similarities to glutamate; thus, it may communicate with the surrounding neurons via the interaction with the NMDA receptor [130]. Indeed, it has been demonstrated that activation of NMDA receptors, whether by glutamate or D2-HG, leads to increased membrane excitation, and thus, epilepsy [131,132]. The epileptogenic activity elicited by IDH mutation on the neighbouring neurons has been recently demonstrated to be triggered by the mTOR pathway, and it can be blocked by adding rapamycin [116]. However, the extracellular concentration of D2-HG can reach the mM range [105,133], while the normal extracellular concentration of glutamate is estimated to be between the nM and µM range [134]. This suggests that the levels of D2-HG in mutant IDH gliomas may also be excitotoxic by leading to an exacerbated or prolonged activation of glutamate receptors.

### 2.5. The Role of IDH Mutation in Tumour Invasion

Glioma cell invasion is one of the main reasons for tumour recurrence, and thus, poor prognosis. The exact mechanisms underlying invasion are yet to be elucidated. Mathematical modelling using data obtained from MRI scans has demonstrated that IDH mutant tumours are less aggressive but more diffuse and invasive than their WT counterparts [135]. To explain this phenomenon, it has been proposed that IDH mutant tumours can modulate their microenvironment by increasing the extracellular levels of D2-HG. High levels of D2-HG might acidify the stroma of the tumour to a degree toxic to normal cells, causing tissue degeneration and space formation for the migration of glioma cells [135]. However, alternative explanations have also emerged, since no evidence of extensive acidosis and tissue damage is observed in most grade II and grade III gliomas, where D2-HG production normally occurs. It has been proposed that glioma cells, especially IDH mutant cells, in which α-KG and NADPH levels are decreased [136], import glutamate from the tumour microenvironment, which is used to produce α-KG in the anaplerotic reaction glutaminolysis. Therefore, in IDH-mutated glioma cells, the neurotransmitter glutamate could act as a chemotactic compound [137].

In colorectal cancer cells, D2-HG was shown to promote motility and invasion of cancer cells. Exogenous administration of D2-HG directly induced epithelial–mesenchymal transition (EMT) by increasing the genetic expression of ZEB1, a master regulator of EMT. Interestingly, D2-HG was applied at 250 µM, while endogenous production of D2-HG by IDH mutation has been known to reach the mM range [133]. This would suggest that this phenomenon may be present even at an enhanced level in human disease. Interestingly, the same study found that elevated levels of D2-HG were present in some colorectal cancer cells and specimens without mutations in IDH. In particular, colorectal cancers with high levels of D2-HG were the samples with an increased frequency of distant organ metastasis [138]. Another study involving glioma cell lines T98 and U87 also showed that exogenous application of D2-HG led to increased cell proliferation and invasion. This study went one step further by introducing the IDH1 R132H mutation via a lentiviral transduction method. These cells—along with non-transduced, D2-HG treated cells—had an increase in protein expression of β-catenin, which is involved in the invasion-associated epithelial–mesenchymal transition [139]. Interestingly, another study utilising the overexpression of IDH1 R132H in U87 and U251 cells demonstrated the opposite effect on cell proliferation and invasion, while also having a negative impact on β-catenin [140]. This contradictory evidence suggests that IDH mutation, subsequent production of D2-HG and cell invasion mechanisms are not linear and may have many other factors involved, which experimental conditions can influence.

Interestingly, in vivo studies in which IDH1 R132H was selectively induced in the brain subventricular zone (SVZ) of the adult mouse demonstrated that SVZ cells gained the ability to proliferate ectopically and spread outside of the SVZ into the surrounding brain regions. This process does not occur in healthy mice [71]. This indicates that IDH1 mutation allows cells to invade the nearby parenchyma in the initial stages of tumour development.

### 2.6. The Emerging Role of IDH3 in Cancer and Beyond

IDH3 is the third member of the IDH family and a vital enzyme of the TCA cycle. As mentioned previously, IDH3 catalyses the irreversible conversion of isocitrate to α-KG, meanwhile reducing NAD^+^ to NADH [141]. NADH, produced during the TCA cycle, is then used to generate ATP by the electron transport chain [142].

Increasing experimental evidence indicates that IDH3 is involved in the epigenetic regulation of the genome. The α-KG is an obligatory co-factor of the α-KG-dependent dioxygenases, a superfamily of enzymes involved in many biological processes, including epigenetic regulation of gene transcription [143]. IDH3, by producing αKG, may modulate the enzymatic function of the α-KG-dependent dioxygenases, consequently influencing the demethylation of DNA and histones. It has been demonstrated that, like several other TCA enzymes, such as pyruvate decarboxylase and pyruvate dehydrogenase, IDH3A localises temporarily to the nucleus during zygotic development, and it contributes to the expression of genes essential for zygotic genome activation and developmental progression [144]. Another recent study demonstrated that mitochondrial metabolites, including TCA cycle intermediates, play critical roles in regulating cell pluripotency. During somatic cell reprogramming to pluripotency acquisition, many TCA cycle enzymes, including Idh3a, localised to the nucleus to induce epigenetic remodelling [145]. Moreover, recently, a non-classical and incomplete TCA cycle was discovered in the nucleus of eukaryotic cells to possibly generate and consume metabolic intermediates used not for energy production but for epigenetic regulation [146].

Loss-of-function and missense mutations in *IDH3A* and *IDH3B* have been linked to inherited retinal diseases (IRDs), which are a leading cause of blindness in children and adults [147,148,149]. Moreover, like many other TCA cycles enzymes whose genetic mutations have been implicated in neurological disorders [150,151,152,153,154,155,156,157,158], a homozygous missense mutation (p.Pro304His) in *IDH3A* has been associated with a severe form of encephalopathy, which causes neurological defects from birth and retinal degeneration [159].

Currently, *IDH3* has not been found mutated in any tumours. In a screen of 47 glioblastoma samples, no *IDH3* mutations were found [160]. However, abnormal expression of IDH3 has been implicated in the cause and development of several cancers [161,162,163,164].

IDH3α is overexpressed in various human cancer cell lines [161] and cancers [164], and in patients affected by human hepatocellular carcinoma (HCC) [164], lung and breast cancers [161] high levels of IDH3α expression were correlated with a poor prognosis. Interestingly, IDH3α was identified as a novel upstream activator of HIF-1α. In vitro data showed that IDH3α overexpression results in an upregulation of HIF1α in both normoxic and hypoxic conditions. By reducing the level of intracellular α-KG, the aberrant expression of IDH3α was found to suppress the hydroxylation of HIF-1α protein and consequently to upregulate HIF-1α stability and transactivating activity. Accordingly, in vivo data showed that overexpression of IDH3α significantly accelerated the growth of HeLa/IDH3α xenograft tumours, while silencing IDH3α was observed to obstruct tumour growth [161]. IDH3 was also found overexpressed in GBM samples, and it was found to promote GBM progression through its essential role in the one-carbon metabolism, which regulates nucleotide production and DNA through its effect on cytosolic serine hydroxymethyltransferase (cSHMT) [163]. Interestingly, cells on the leading edge of the tumour had increased levels of IDH3α expression compared to those in the centre of the tumour.

Recently, IDH3α has been found to promote the progression of HCC by inducing expression of metastasis-associated 1 (*MTA1*), an oncogene regulating cancer progression and metastasis [164]. The aberrant levels of IDH3α in HCC cells have been found to promote epithelial–mesenchymal transition by inducing MTA1 expression, thereby increasing cell migration and invasion [164].

In contrast, it was found that the downregulation of IDH3α promotes the transformation of fibroblasts into cancer-associated fibroblasts by upregulating HIF-1α, which in turn causes a switch from oxidative phosphorylation to glycolysis [162].

All these studies demonstrated that although the role of IDH3 in cancer is still unclear, the impact of IDH3 on tumour initiation and progression is definitively context-dependent. Indeed, in some cell types, aberrant expression of IDH3 is advantageous, whereas in others, downmodulation of IDH3 is important to promote cell transformation. The different consequences of IDH3α aberrant expression for various tissue-specific cells could be explained by the diverse metabolic requirements of each tissue during malignant transformation.

## 3. Novel Therapeutic Options for *IDH* Mutant Glioma

### 3.1. Direct Inhibition of Mutant IDH

Over the last decade, direct inhibitors of mutant IDH have rapidly developed to decrease the levels of D2-HG or 2-HG in IDH mutant tumours. AGI-5198 has been reported as the first novel, synthetic, direct enzyme inhibitor of IDH [165]. Celgene Corporation and Agios Pharmaceuticals was the first collaboration to generate IDH mutant inhibitors via a high-throughput screening campaign. AGI-5198 is a phenyl-arginine-based compound able to block the generation of 2-HG produced by IDH mutant cells by 90% and to impair the growth of IDH mutant xenograft in vivo [166]. As reported by Rohle and colleagues, AGI-5198 showed efficacy in patient-derived glioma xenograft models, and the drug also induced the differentiation of glioma cells and reduced cell proliferation and histone methylation. However, it is worth noting that the global DNA methylation leading to the glioma CIMP phenotype was notably unchanged after the administration of the drug [165]. A study using IDH1 mutant knock-in human astrocyte models later suggested that the IDH1 inhibition has a small effective time frame because the IDH1 mutation is likely to change from a “driver” to a “passenger” mutation [167]. Despite the reduced proliferation and histone modification four days after the oncogenic insult, the drug could not block or reverse the IDH1 mutation phenotypic changes.

Further, work by Tateishi and colleagues showed that treating patient-derived IDH1^R132H^ glioma tumour spheres with the S-enantiomer of the AGI-5198 does not block cell proliferation despite the decrease in 2-HG levels, and in vivo data using an orthotopic xenograft glioblastoma IDH mutant model showed similar survival of mice treated with the AGI-5198 S-enantiomer compared to the mice treated with vehicle [168]. These mixed results in glioma models can be explained by the fact that IDH mutations are an early event in gliomagenesis and an oncogenic driver in LGGs; however, during tumour evolution, IDH mutant gliomas acquire a plethora of other subsequent mutations, which might render gliomas, and especially high-grade gliomas, less susceptible IDH mutant inhibitors. Another hypothesis is that an epigenetic memory might be indelible by the IDH mutation enzymatic inhibition. Thus, a combinatorial approach to treatment may be needed. In addition, the fact that AGI-5198 is rapidly metabolised and cleared leads to the conclusion that it might be a poor candidate for use in clinical trials [166].

AG-120 (ivosidenib), AG-881 (vorasidenib) and AG-221 (enasidenib) are the second generation of selective, reversible drug inhibitors produced, which the F.D.A. approves for the treatment of acute myeloid leukaemia [169,170]. AG-221 was the first selective IDH mutant inhibitor drug approved by the F.D.A. in 2017 (Table 1). Initially, a drug precursor was chosen via high-throughput screening as the most prevalent form of IDH2 in AML, IDH2^R140Q^ [37,171]. The precursor could bind to an allosteric site of IDH2 within the heterodimer interface of the enzyme, much like its predecessor AG-6780 [172]. However, it was shown via X-ray crystallography that the heterodimer interface binding forces IDH2^R140Q^ to adopt a non-catalytic open conformation leading to inhibition consistent with the functional changes described in IDH1^R123H^ inhibition [173]. Subsequent modifications optimised the potency, solubility, clearance and bioavailability of the drug, leading to the development of AG-221 [171]. AG-221 showed time-dependent potency, leading to the reduction in 2-HG in biochemical assays [171,174] with subsequent 2-HG depletion in transgenic IDH2^R14OQ^ TF-1 erythroleukemia cells and in patient-derived primary AML cells expressing either IDH2^R140Q^ or IDH2^R172K^ [171]. In vivo, within 20 days, the mouse modelwith patient-derived IDH2^R140Q^ AML cells produced cell-specific markers of differentiation, such as CD11b, CD14 and 15, with a reduction in the CD17 immature marker, and showed an increase in survival. No information exists yet regarding using AG-221’s penetrance through the BBB. In addition, the fact that IDH2 mutations are less abundant in gliomas did not lead to many clinical studies on glial tumours. There is a single trial of dose escalation for gliomas (NCT02273739, Table 1, and other IDH mutant tumours, which started in 2014). The drug showed inhibitory effects for these tumours, but appropriate dosing was an issue. 

Celgene/Agios developed AG-120 (Ivosidenib) to optimise the AGI-5198 for human therapeutic applications [189]. Broad structure activity profiling led to the replacement of the cyclohexyl moieties with the fluorinated cycloalkyl groups, preventing oxidation by the liver [166]; moreover, the addition of the pyrimidine ring allowed for better potency, leading to the drug called AG-14100. However, AG-14100 was a potent liver enzyme inducer. Subsequent modification led to AG-120 with good potency and clearance. AG-120 is a specific allosteric, reversible inhibitor of mutant-IDH1, competing with e magnesium, an essential co-factor for the enzyme, preventing the formation of the catalytic site [190]. AG-120 inhibited many IDH^R132H^ mutant cell lines selectively [191]. A significant disadvantage is the low penetrance (4.1%) in mouse models with intact BBB, which may be increased with a disrupted BBB in glioma models [165]. More animal studies are needed to evaluate its penetrance.

Regarding mouse models, a dose-dependent 2-HG depletion has been observed across cell types, including an IDH1^R132H^ mutant glioma xenograft mouse model [165,192,193]. In addition, the same articles reported that AG-120 could modulate some oncogenic properties of cancer cells, such as inducing differentiation in AML myeloblasts and inhibiting invasion and migration in chondrosarcoma cell lines [192,193,194]. Afterwards, the drug was approved by the F.D.A. for refractory/relapsed AML and is currently under investigation in two clinical glioma trials evaluating its efficacy and safety profile (Table 1). In 2015, the phase 1 data on the safety of AG-120 in patients with advanced gliomas among other solid tumours showed that the drug is well tolerated with a promising pharmacokinetic profile [175]. In the trial, AG-120 was initially administered in a dose-escalated manner over 28 days ranging from 100 mg to 1200 mg. The maximum dose resulting in a more significant D2-HG reduction was 500 mg. Therefore, 500 mg was chosen for the trial’s dose expansion phase for enhancing and non-enhancing IDH mutated gliomas [176]. The latest data showed that the AG-120 optimal dosing regime of 500 mg once a day led to a 98% reduction in the 2-HG in cholangiocarcinoma and chondrosarcoma patients [176]. However, the plasma 2-HG levels in glioma did not seem to be a robust pharmacokinetic marker. Therefore, a second multicentre trial is underway on recurrent LGG non-enhancing gliomas with IDH mutation using an AG-120 and AG-881. The primary outcome is to compare the D2-HG concentrations in surgically removed tumours, which were treated versus not treated with the drug inhibitors [195]. Clinical safety, dosage, tolerance and pharmacokinetics will also be studied. This safe feasibility trial will provide appropriate dosing of AG-120 for future randomised studies. It is worth noting that the new inhibitors exhibit a good CSF–plasma ratio [166]. On the contrary, the main issue with the most common chemotherapeutic agent for HGGS, temozolomide (TMZ), was whether the amount of the drug reaching the tumour might not be enough to eliminate the remaining cells. Importantly, no serious adverse events have been reported using AG-120.

AG-881, also named Vorasidenib, is the only oral pan-inhibitor for IDH1 and IDH2 mutations developed by Celgene and Agios Pharmaceuticals [129,177,196]. AG-881 was shown to bind the allosteric pocket at the dimer interface, leading to steroid hindrance and subsequent conformational change to an open and inactive state [196]. AG-881 achieved more efficient inhibition in vitro with shorter incubation periods for the IDH1^R132H^ mutation [196]. Experiments showed the inhibition of 2-HG formation following 1 h of incubation in genetically engineered patient-derived TS603 glioma cell lines expressing IDH1^R132C, G, H, L, S^ [197]. In TF-1 and U87 cells engineered to express IDH2^R140Q^ and IDH2^R172K^ mutant enzymes, 2-HG inhibition was also shown, and cellular differentiation was increased [177]. AG-881 was shown to penetrate the BBB in healthy rodents, potentially holding promise for use in humans. Indeed, AG-881 is currently under a phase 1, open-label, dose-escalation and expansion trial for the safety and pharmacokinetic properties to be investigated in both IDH1 and IDH2 mutated gliomas [170]. According to the data presented in 2021, the drug shows a favourable pharmacokinetic profile with no severe adverse effects and an increase in survival to 24 months in 60% of the patients [170]. The median progression-free survival was 36.8 months (95% confidence interval (CI), 11.2–40.8) for patients with non-enhancing glioma and 3.6 months (95% CI, 1.8–6.5) for patients with enhancing glioma. Previously, a 25–300 mg dose range was tested in the dose-escalation arm and 10 or 50 mg in the dose-expansion arm [170]. Doses >100 mg lead to toxicity. The most advanced clinical trial design is a phase 3, multicentre, randomised, double-blind, placebo-controlled study of AG-881 in subjects with residual or recurrent grade 2 glioma with an IDH1 or IDH2 mutation (Table 1) [178]. Approximately 366 participants will be randomised 1:1 to receive orally administered AG-881 50 mg QD or placebo (Table 1). There is now (March 2023) a press release awaiting official results to be published.

BAY-1436032 is a relatively new species, non-competitive IDH1m inhibitor with adequate pre-clinical results in AML and glioma models (Table 1). This compound is an allosteric IDH1 enzyme inhibitor, which binds to the dimer interface of mutant IDH [180]. BAY-1436032 was selected from 3 million compounds based on IDH enzymatic activity with IC50 ranging from 0.6 to 17 micrometres [180]. The drug showed equivocal inhibition of all IDH1^R132^ mutants in AML human-derived cell lines and genetically engineered lines of solid tumours with reduced proliferation and induction of differentiation. In astrocytoma xenograft IDH1^R132H^ mouse models, the drug effectively penetrated the BBB with prolonged survival. Two ongoing AML and solid tumour trials are reviewed here [198]. The glioma trial is an open-label, non-randomised, multicentre phase I study to determine the maximum tolerated or recommended phase II dose of oral mutant IDH1 inhibitor BAY1436032 and to characterise its safety, tolerability, pharmacokinetics and preliminary pharmacodynamic and anti-tumour activity in patients with IDH1^R132X^ mutant advanced solid tumours, including gliomas [181] (Table 1). In dose escalation, 29 subjects with various tumour types across doses in the dose escalation ranging from 150 to 1500 mg demonstrated a 76% reduction in D2-HG levels. A dose of 1500 mg was chosen for the dose expansion phase. Thirty-five glioma patients showed an 11% objective response rate and 43% disease stability [181]. The full results are to be published. Another new inhibitor, the MRK-A, achieved a robust intracranial 2-HG inhibition in the orthotopic mouse brain tumour models generated using BT142 and GB10 glioma cells, where IDH mutation had naturally occurred. However, even with near complete inhibition of intratumoural 2-HG production, not all IDH mutant glioma models responded to treatment, but only BT142 displayed significant tumour growth inhibition resulting in a measurable survival benefit. Pronounced differences in the gene expression patterns between BT142 and GB10 tumours were also observed following MKR-A treatment [199].

IDH-305 is an allosteric non-competitive inhibitor of IDH1^R132C, H^ developed by Novartis. IDH-305 stabilises the enzyme via an inactive conformational change [182], effectively reducing 2-HG levels with substantial blood–brain barrier penetrance in murine models. A current phase 1, single group assignment, open-label for advanced malignancies harbouring IDH^R132H^ mutations trial has just been published (NCT02381886, Table 1) [183]. In total, 35/41 patients demonstrated target engagement with reduced 2-HG concentration at all doses, 75–750 mg twice daily. Complete remission (CR) or CR with incomplete count recovery occurred in 10/37 (27%) patients with AML and 1/4 patients with myelodysplastic syndrome. Adverse events (AEs) suspected to be related to the study drug were reported in 53.7% of patients. A few new IDH1-specific inhibitors exist, such as the FT-2102 a competitive inhibitor of IDH^R132C^, in clinical trial monotherapy [186] (Table 1). The Ds 1001b direct IDH1^R132X^ inhibitor with evidence of penetrance through the BBB in humans and xenograft mouse models is also highly promising, with the phase 1, single group assignment trial showing tolerance at 1400 mg with a progression to phase 2 trial [188] (Table 1). Phase 1 showed that twice daily oral administration resulted in anti-tumour activity in patients with recurrent/progressive IDH1-mutated glioma. A phase II study of DS-1001 in patients with chemotherapy- and radiotherapy-naïve IDH1 mutant WHO grade 2 gliomas is ongoing to verify the efficacy of DS-1001 as a single agent (NCT04458272).

A few new inhibitors are noted in the table, with some promising results in other malignancies, such as the AML outlined in another comprehensive review [198]. Despite the positive results, the success of IDH mutant inhibitors is found to have a plethora of limitations in IDH mutant gliomas. Although AGI-5198 reduces neomorphic activity, a study showed that it does not alleviate the DNA and histone hypermethylation phenotype, since histone methylation was high [167]. Further, Sulkowski and colleagues demonstrated that AGI-5198 prevents DNA damage in cancer cells, concluding that this might allow for resistance to DNA damage agents, such as the current chemo- and radio-therapeutic options [200]. Another study also confirmed this, suspecting that IDH1 mutated cells under the action of AGI-5198 gain radioprotective abilities [179]. Currently, other novel molecular inhibitors are being tested. Those can be combined with IDH inhibitors to overcome each possible drawback through umbrella and ongoing platform trials, such as the Tessa Jowell BRAIN MATRIX Platform (TJBM) [201].

### 3.2. IDH Vaccine

IDH mutations are an early event in gliomagenesis [202] and are present in recurrent gliomas [203,204]. This makes mutant IDH an excellent target for potential immune therapies. Initial attempts to produce vaccines against mutant IDH were performed by Schumacher et al., where they inoculated mice with a twenty-amino-acid peptide spanning part of the mutated catalytic pocket of the IDH enzyme. This resulted in mice mounting a robust immune response from CD4^+^ T-helper cells specific to the IDH mutation. The growth of subcutaneous sarcomas carrying either IDH1 WT or IDH1^R132H^ was investigated following preventative vaccination: tumours positive for IDH1^R132H^ grew more slowly than those expressing the WT protein [205], showing the efficacy of the anti-tumour therapy. A direct follow-up to this initial vaccine research was the NOA16 trial, a first-in-human, single-arm phase 1 trial, in which newly diagnosed patients with grades 3 and 4 IDH1^R132H^ astrocytomas were recruited [206]. IDH mutation confers a survival advantage to patients diagnosed with grade 3 or 4 gliomas and is a unique target for immune therapy [207,208]. The trial comprised eight vaccinations, and 90.6% of participants experienced non-severe adverse effects. A more significant percentage of trial participants experienced pseudo-progression—a phenomenon whereby a tumour appears to have increased in size when viewed via imaging, but where no such growth has occurred—compared to a molecularly matched control group, which was indicative of an immune response induced by the IDH1^R132H^ vaccine. However, the authors concluded that the pseudo-progression seen may have been a delayed response from previous rounds of radiotherapy, as they excluded patients presenting with pseudo-progression from the trial [206].

In summary, research indicates that mutant IDH is a potential target for immunotherapy. The phase 1 NOA16 trial shows that the vaccine is well tolerated and safe, thus paving the way for phase 2 trials to assess the vaccine’s effectiveness. Patients included in the NOA16 trial were diagnosed with either WHO grade 3 or 4 astrocytomas, leaving the door open for IDH vaccination research in patients affected by oligodendroglioma to understand whether IDH vaccination can benefit this group. The IDH^R132H^ vaccine has made promising steps towards becoming a novel therapeutic option for glioma. However, mutations in either IDH1 or IDH2 occur in many other cancers, which can be a possible susceptible target by IDH mutant vaccination. Glioma most commonly contains the IDH^R132H^ point mutation [209]; however, other cancers mainly express other IDH variants, including IDH1^R132C^ and IDH1^R132G^, but also mutations in IDH2, commonly IDH2^R140Q^ and IDH2^R172K^ [210]. This means that many potential targets for IDH-specific vaccination treatments across different cancers exist.

### 3.3. Modulating Epigenetic Alterations in IDH Mutant Gliomas

As previously mentioned, the IDH mutation leads to histone and DNA hypermethylation patterns [40,63,64,96,97,98,100,102,103,104,211,212].

The hypermethylation phenotype might lead to oncogenic activities in IDH mutant cells. Thus, intervening with the epigenetic changes has been postulated as a potential therapeutic option for IDH mutant gliomas. Flavahan and colleagues have demonstrated that glioma CpG island methylator phenotype (G-CIMP) is linked to hypermethylation at sites for cohesion and CCCTC-binding factor (CTCF), leading to the reduced affinity of this protein [213]. The CTFC-reduced binding affinity allows for the enhancer-mediated expression of PDGFR-A. PDGFR-A is a known mitogen linked to glioma genesis [214]. By administrating a demethylating agent, they showed that the CTCF binding is partially restored, and the PDGFR-A expression is reduced. Another study has also documented the notion that inhibiting hypermethylation might be beneficial. Decitabine, a DNA methyltransferase inhibitor, could suppress the proliferation, both in vitro and in vivo, of IDH mutant glioma cells [215]. Concomitantly, 5-azacytidine, an analogue that controls the DNA methyltransferase activity, has reduced the proliferation of the IDH-mutated xenograft glioma model [216]. However, epigenetic changes are only a piece of the puzzle. Combinatorial therapies might be needed to tackle the oncogenic potential induced by IDH mutations.

### 3.4. Inhibiting DNA Repair

DNA is susceptible to various damaging agents from within the body, such as replication errors and toxins, and from outside the body, such as UV and ionising radiation [217]. Because of this, we have evolved an impressive array of mechanisms to counteract this damage, including homologous repair (HR) and non-homologous end joining (NHEJ) [218].

As mentioned, the tumourigenic effects of D2-HG are proposed to derive from modulating the α-KG-dependent dioxygenases, many of which are involved in histone and DNA methylation, but also enzymes playing critical roles in the DNA damage response, such as alkB homologue (ALKBH) enzyme [219,220]. Wang et al. found that glioma cells expressing mutant IDH reduced repair kinetics, accumulated more DNA damage and were sensitive to alkylating agents. The overexpression of ALKBH2 or AKLBH3 could reverse the sensitisation to alkylating agents. These data indicate that alkylating agents may represent a valuable therapeutic option for treating IDH-mutated cancer patients [219].

D2-HG can also compromise the HR of DNA damage response [200]. Sulkowsky et al. found that applying a 2-HG analogue—2R-octyl-α-hydroxyglutarate—to IDH1 WT cells caused an increase in double-strand breaks, suggesting that D2-HG was able to inhibit the HR pathway [200]. Alongside HR, another pathway of DNA repair relies on the poly (ADP-ribose) polymerase (PARP) family of enzymes. PARP enzymes play a role in the repair of single-strand breaks, where they perform base-excision and nucleotide-excision repair [221]. Interestingly, when the HR pathway is inhibited in mutant IDH1 cell lines due to the D2-HG accumulation, PARP enzymes become a point of failure in the system. Specifically, inhibiting PARP in these IDH1 mutant cells leads to the selective death of mutant but not WT cells.

Conversely, inhibiting the neomorphic activity of mutant IDH1 with specific small molecule inhibitors reversed the deficiency in HR, with the number of double-strand breaks almost returning to that seen in IDH1 WT cells. Interestingly, the PARP inhibitor sensitivity induced by mutant IDH1 is present and functional in both patient-derived AML and glioma cells [200]. Taken together, these data indicate that IDH1 mutant cells are sensitive to the activity of PARP inhibitors, and this has the potential to be exploited in IDH mutant malignancies. In another study, it was demonstrated that depleting NAD^+^, which is needed for PARP during TMZ-induced BER, using GMX1778 and inhibiting Nicotinamide phosphoribosyltransferase (NAMPT) using FK866, eliminates the remaining repair activity of PARP [168,222]. This induces a specific metabolic stress response to TMZ-induced DNA damage and improves the duration of the therapy response.

Additionally, a currently active phase II/III trial—NCT02152982—investigates whether the combination of TMZ and PARP inhibitor veliparib is more effective at treating newly diagnosed glioblastoma than TMZ alone. The recruited patients must have MGMT promoter methylation, as this indicates that tumours are sensitive to TMZ and that the added inhibition of PARP DNA repair would further exacerbate DNA damage, leading to a higher rate of cell death [223]. At the time of writing, this trial reached its primary completion date on 1 December 2021, with the estimated study completion date being 15 December 2024 (https://clinicaltrials.gov/ct2/show/study/NCT02152982, accessed on 20 March 2023).

The telomere lengthening is a mechanism cancer cells employ to maintain their growth over time [224]. Low-grade astrocytomas and secondary glioblastomas frequently have a loss of the *TERT* gene, which encodes for the telomerase, the enzyme responsible for maintaining the structure of telomeres Consequently, cells would be stunted in their growth, as they would not be able to maintain the length of their telomeres. However, *TERT* loss is usually accompanied by the deletion of *ATRX* and *IDH1* mutation, which, when combined in cells, is associated with alternative lengthening of telomeres (ALT) [225]. ALT is a poorly understood mechanism related to homologous recombination, which allows tumour cells to continue growing while protecting their telomeres from replicative shortening [226]. In low-grade gliomas, it may be possible for IDH inhibitors to ameliorate the ALT phenotype, since the triad of loss of TERT, loss of ATRX and IDH mutation are required to produce the ALT phenotype [225]. However, given that IDH mutation is such an early event in gliomagenesis, administering inhibitors in time may not be clinically realistic to prevent tumour formation. In light of this, there have been studies into inhibitors of the ALT mechanism itself, including an inhibitor of ATR kinase, which leads to an increase in the fragility of telomeres [226]. In conclusion, genomic instability and glioma metabolism are interrelated and thus offer a unique area to explore therapeutic strategies.

### 3.5. Inhibiting Metabolic Pathways

By producing D2-HG, IDH mutant enzyme drains some of TCA cycle intermediates. Therefore, anaplerotic pathways are recruited to compensate for the loss of TCA cycle metabolites, such as α-KG. Understanding and delineating the new metabolic alterations will allow us to evaluate potential druggable targets as novel therapeutic options.

NAD is an indispensable co-factor in the cell, as carrying H^+^ ions is important for the electron transport and metabolism of redox reactions, including glycolysis, TCA cycle and serine biosynthesis. Furthermore, by affecting the activity of several enzymes, NAD functions as an essential element for various signalling pathways. In the cell, NAD is produced from two different pathways, the de novo pathway or the ubiquitous salvage pathway (classical and alternative) [227]. Even though most cells can synthesise NAD through the de novo pathway, the primary source of NAD is supplied by salvage pathways. Nicotinamide phosphoribosyltransferase (NAMPT) and nicotinate phosphoribosyltransferase (NAPRT) are critical enzymes for the NAD salvage biosynthesis pathway, and they synthesise NAD from nicotinamide (NAM; classical pathway) or nicotinic acid (NA; alternative pathway), respectively [227].

In IDH mutant glioma cells, the alternative salvage pathway of NAD is reduced due to the epigenetic silencing of the *NAPRT1* gene [168]. Thus, inhibiting the classical salvage pathway in these cells using inhibitors targeted to the NAMPT enzyme resulted in a lack of metabolic substrates and a biochemical crisis, which activated the energy sensor AMPK to initiate autophagy and the consequent cytotoxicity. In contrast, to maintain the energy supply of NAD during metabolic stress, the IDH wild-type cells increased the expression of NAPRT1 or access to alternative metabolic pathways. The study concluded that IDH1 mutation could make cells NAD-dependent gene [168]. Thus, it is a rational therapeutic target in metabolic pathways.

Moreover, in IDH mutant cells, glutaminolysis is the major pathway of metabolic compensation due to a lack of isocitrate [78]. Thus, targeting glutamine/ate metabolism might deplete the energy sources and thus inhibit major anabolic functions of the IDH mutant cell. For instance, bis-2-[5-9phenylacetamide)-1,3,4-thiadiazol-2y]ethyl sulfide (BPTES) has been shown to block glutaminase and thus hamper the glutamate metabolism and reduce proliferation and growth in IDH mutant AML cell [79,228]. Further, another drug called Zaprinast was able to block glutaminase and reduce the proliferation of IDH-mutated glioma cells [229]. Moreover, another glutamine inhibitor, telaglenastat (CD-83), was shown to cause reduced D2-HG production and induce glioma differentiation [230]. A phase 1 RCT, combining TMZ, radiotherapy and the glutaminase blocker CB-83, is about to start recruiting patients affected by IDH-mutated diffuse and anaplastic astrocytomas [231]. By suppressing glutaminolysis, tumour growth might cease, and differentiation might occur.

### 3.6. Modulating Redox Homeostasis

ROS are predominantly elevated in IDH-mutated tumours [232]. It was found that glutamine/ate and glutathione are reduced in IDH-mutated glioma cells compared to adjacent normal tissues. D2-HG is negatively correlated with the levels of glutathione. Therefore, glutathione may be essential for IDH mutant maintenance of redox homeostasis [233]. Increased consumption, and thus, reduction in glutathione suggests the increased burden of ROS scavenging. Understanding these relationships will allow for therapies able to intervene in redox homeostasis. Limiting ROS scavenging, driven by glutathione, could be an add-on therapy to the existing or under-trial therapies. As previously mentioned, CB-839 can lead to blockage of glutamine metabolism and thus impair redox homeostasis and sensitisation to radiotherapy [234].

### 3.7. Differentiation Therapy

One main consequence of IDH mutation is its ability to block normal cellular differentiation. Studies performed in various IDH mutant malignancies have demonstrated that mutant IDH can halt normal cell differentiation, causing an expansion of progenitor cells, which is an important step in cancer development [40,235,236,237,238]. Inhibition of mutant IDH with small molecule inhibitors can release the differentiation block and possibly hinder tumour growth.

In the glioma context, Rohle et al. demonstrated that inhibition of mutant IDH by using a small IDH mutant selective inhibitor, AGI-5198, hampered the in vitro and in vivo growth of an oligodendroglioma cell line (TS603), harbouring an endogenous IDH1-R132H mutation. Treatment of mice xenografts with AGI-5198 caused an upregulation of several genes involved in glial differentiation, while the in vitro treatment of TS603 cells with AGI-5198 showed a loss of repressive histone marks at the promoters of astrocytic marker genes. This study demonstrated that, at least in this model, targeting mutant IDH1 can impair glioma development in vivo, which is related to changes in cellular differentiation [165].

More recently, an undifferentiated glioma cell line, BT142, which expresses endogenous IDH mutation and has an undifferentiated cell state was treated with a mutant IDH inhibitor (MRK-A). MRK-A caused decreased expression of some key stem cell markers in vitro indicating an onset of differentiation. Treatment of mice bearing orthotopic BT142 tumours with MRK-A showed a reduction in the 5mC DNA signature, indicating a release of the hypermethylation phenotype seen in IDH mutant tumours. The treatment with MRK-A in vivo showed a significant upregulation of 245 genes, which is in line with removing the hypermethylated phenotype. Mice that received MRK-A treatment showed an extended survival and reduced Ki67 staining compared to their vehicle-treated control group [199]. This study has begun to explain how differentiation therapy may benefit the treatment of IDH mutant brain tumours, as it removes the 2-HG-dependent hypermethylation phenotype and alters the expression of a wide range of genes, which may enable differentiation of the cells. Further work is needed to understand whether there are any long-term effects of releasing these genes from their suppression via methylation and whether differentiation therapy can be combined with other treatment modalities.

IDH inhibitors as differentiation therapy have also been shown in haematological malignancies, such as acute myeloid leukaemia (AML). Wang et al. developed a specific inhibitor for the most commonly occurring IDH mutation in AML, IDH2-R140Q, AGI-6780. Treatment with this inhibitor of an erythroleukemia cell line, which ectopically expressed IDH2-R140Q, decreased the levels of 2-HG and released these cells from the block to differentiation. Similar studies on IDH2-mutated primary human AML cells showed a reduction in 2-HG levels and an increased differentiation of mutated AML cells [172]. These data suggest that mutant IDH causes a block to differentiation, which is released upon treatment with an inhibitor.

More recent studies have shown that AG-120 (ivosidenib) treatment decreased intracellular levels of 2-HG and induced differentiation in models of *IDH1*-mutated tumours [166,239]. Recently, AG-120 has been licensed for use in relapsed or refractory AML [240]. AG-120 was tested in phase I clinical trial, where it was shown to induce a substantial remission in patients with mutant IDH relapsed or refractory AML [169].

The simultaneous use of an IDH1 inhibitor, BAY1436032, and a DNA methyltransferase inhibitor, azacitidine, has proven to be a novel, clinically relevant drug combination in treating AML. The frequency of leukaemia stem cells (LSCs) in PDX models was analysed, and it was found that they were decreased 4.1-fold with azacytidine alone and 117-fold with BAY1436032 alone. However, the frequency of LSCs was reduced by 470-fold when the drugs were given sequentially and by 33,150-fold when given simultaneously. These data show that it is important to continue evaluating the synergistic effects of different drugs. Similarly, azacitidine has also been shown to reverse the differentiation block caused by mutations in IDH2 in chondrosarcoma cell lines [241]. This raises the possibility that IDH mutant chondrosarcoma could be targeted by dual therapy equivalent to AML outlined above.

The use of ivosedenib or enasidenib in treating mutant IDH AML presents a risk of the patient developing differentiation syndrome (DS) [242]. While the exact cause of DS has yet to be established, it is currently thought to arise from the sudden release of cytokines from myeloid cells, as they are forced to differentiate [243]. With this in mind, care must be taken when studying the effects of differentiation therapy in treating brain tumours, as the patient may have unintended consequences and symptoms.

## 4. Conclusions and Future Direction

In last decades, mutations identified in *IDH1* and *IDH2*, and a few other metabolic genes, such as succinate dehydrogenase (*SDH*) and fumarate hydratase (*FH*), have provided compelling evidence for metabolic alteration in human cancer development and progression. However, the mechanism from altered metabolism to tumourigenicity is still unclear.

Increasing genetic evidence has indicated that mutant IDH1 is a driver of gliomagenesis and a rational therapeutic target, leading to a massive effort to design and test specific IDH mutant inhibitors. Although targeted inhibitors of mutant IDH have shown promise in IDH mutant acute leukaemia patients, no benefits have been obtained for patients affected by gliomas, despite the reduction in D2-HG after treatment with inhibitors. Here, we outline key metabolic alterations, epigenetic changes and redox imbalance, which might not be uniform across IDH mutant gliomas and ongoing targeted therapeutic options, with several clinical trials specific to IDH inhibition recently published. However, more light needs to be shed on the possible oncogenic role of D2-HG in IDH mutant malignancies, and more research needs to be focused on the metabolic, cellular and molecular consequences associated with IDH mutations in addition to the accumulation of D2-HG. New IDH1 mutant glioma models are needed to examine the spectrum of responses to treatment, which may be observed clinically following the administration of IDH1 mutant inhibitors. Considering that current therapies fail to demonstrate improvement in glioma treatments, IDH-induced biochemical alterations should be adequately understood and assessed as potential targets.

## Figures and Tables

**Figure 1 cancers-15-02883-f001:**
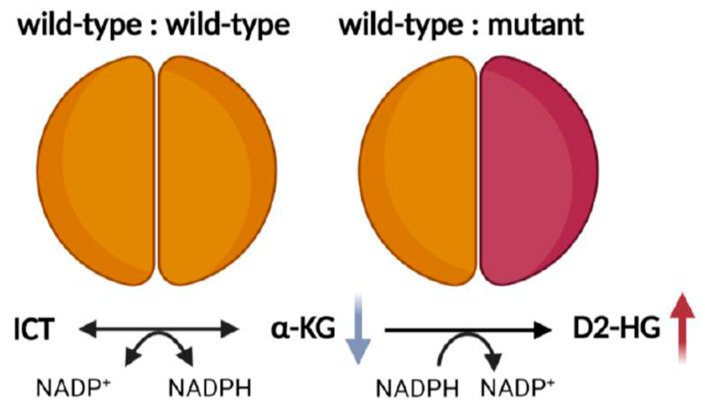
The neomorphic activity of mutant IDH. IDH1 and 2 are homodimeric enzymes, and in the wild-type form, each monomer converts isocitrate to α-ketoglutarate by reducing NADP^+^ to NADPH. Mutant IDH is instead composed of a wild-type and a mutant monomer. Consequently, the wild-type part of the dimer leads to the conversion of isocitrate to α-KG, producing NADPH, whereas the mutant monomer is endowed with neomorphic activity and irreversibly converts α-KG to the D2-enantiomer of hydroxyglutarate (D2-HG), by using NADPH as a co-factor. As a result, in IDH mutant cells, D2-HG accumulates, while α-KG decreases. ICT, isocitrate; α-KG, α-ketoglutarate; D2-HG, hydroxyglutarate. Created with BioRender.com.

**Figure 2 cancers-15-02883-f002:**
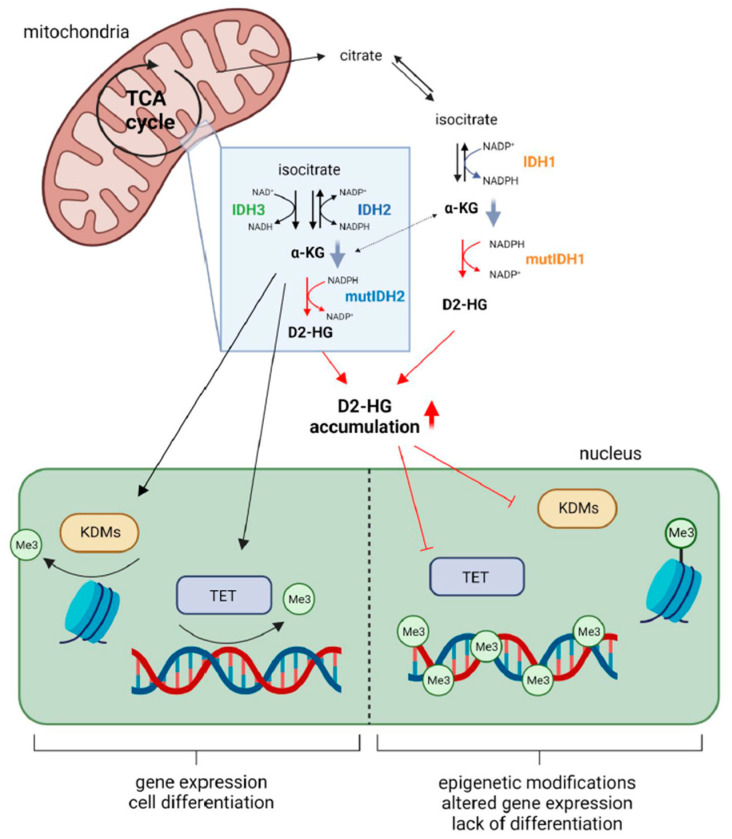
The role of mutant or wild-type IDH and their downstream effects. IDH1 (cytoplasmic), IDH2 and 3 (mitochondrial) enzymes convert isocitrate to α-ketoglutarate (α-KG). Under normal circumstances (represented by the black arrows), α-KG acts directly on α-KG-dependent dioxygenase enzymes, such as Tet methylcytosine dioxygenases (TETs) and histone lysine demethylases (KDMs), within the nucleus to control the epigenetic landscape and regulate gene expression. Mutation of IDH1 or 2 leads to neomorphic activity, which converts α-KG to D2-hydroxyglutarate (D2-HG). D2-HG accumulates within the cell and competes with α-KG to inhibit TETs and KDMs. This causes an accumulation of methyl groups on DNA and histones, which leads to changes in gene expression and causes the affected cells to de-differentiate. Mutant IDH also leads to reduced levels of α-KG [65,66,67,68,69,70,71], which may also contribute to inhibit the α-KG-dependent dioxygenases [63]. α-KG, α-ketoglutarate; D2-HG, D2-hydroxyglutarate; IDH, isocitrate dehydrogenase; KDMs, histone lysine demethylases; TET, ten-eleven translocation; NADP+, oxidised nicotinamide adenine dinucleotide phosphate; NADPH, reduced nicotinamide adenine dinucleotide phosphate; TCA, tricarboxylic acid. Created with BioRender.com.

**Figure 3 cancers-15-02883-f003:**
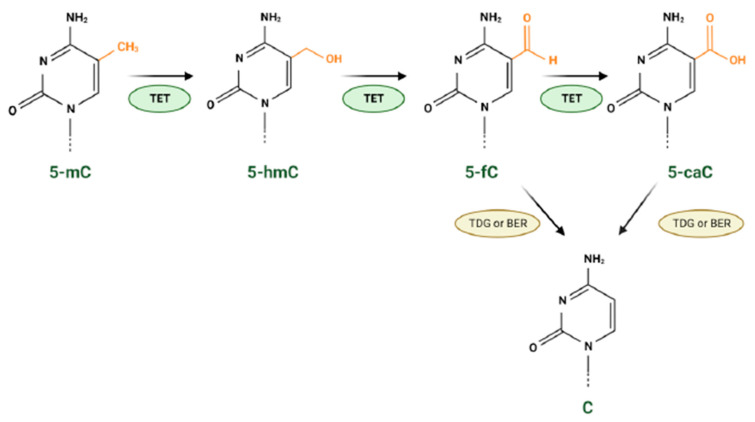
DNA demethylation process catalysed by TET enzymes. During the process of DNA demethylation, TETs oxidise the methyl group of the 5-methylcitosine (5-mC) to 5-hydroxymethylcytosine (5-hmC), and can further catalyse oxidation of 5-hmC to 5-formylcytosine (5-fC) and 5-carboxylcytosine (5-caC). The bases are subsequently recognised and excised by the enzyme thymine DNA glycosylase (TDG) and substituted by an unmodified cytosine (C) by base excision repair (BER). The reactions catalysed by TETs are dependent on α-KG, iron and oxygen. Created with BioRender.com.

**Table 1 cancers-15-02883-t001:** List of selected mutant IDH1 and IDH2 inhibitors.

Drug Name	Target	Mechanism of Action	Advantages	Disadvantages	Current Clinical Trial in Gliomas (s)	Sponsor (s)
AG 120 Ivosidenib (F.D.A. Approved) [169,175,176]	IDH1 R132 C, H, G, S, L	Reversible, allosteric competitive inhibitor	F.D.A. approval was granted in 2018 for relapsed/refractory acute myeloid leukaemia, and in 2019 for newly diagnosed. In 2021, F.D.A received application for the drug to be assessed for cholangiocarcinoma. Currently under investigation in several clinical trials in haematological malignancies.	It is unknown whether it penetrates the BBB, with a 4.1% penetrance in a rat model.	NCT02073994: Phase 1, multicentre, single group assignment, open-label, dose-escalation/safety and clinical activity trial of oral administration for solid tumour, including gliomas. A total of 170 patients to be recruited by 2022. NCT03343197: A phase 1, randomised, multicentre, controlled, open-label, parallel assignment, the perioperative study of AG120 and AG881in patients with non-enhancing IDH1 mutant glioma, both grade II and III. A total of 45 patients estimated.	Agios/Celgene
AG 221 Enasidenib (F.D.A. Approved) [174]	IDH2 R140Q, R172K	Allosteric non-competitive inhibitor	F.D.A. approval was granted in 2017 for relapsed/refractory acute myeloid. Currently under investigation in many clinical trials in haematological malignancies.	No information on penetrance through the BBB.	NCT02273739: A phase1/2 multicentre, open-label, dose-escalation trial for solid tumours, including gliomas. The trial was completed with 21 participants.	Agios/Celgene
AG 881 [170,177,178]	Pan Inhibitor IDH1&2	Allosteric non-competitive inhibitor	Penetrance of the BBB in a rat model. Pan inhibitor with an advanced clinical trial design showing good tolerability and safety profile.		NCT02481154: Phase 1, multicentre, open-label, single group assignment, dose-escalation/safety and clinical activity trial of oral administration for gliomas with IDH1 or IDH2 mutation. The trial was completed in 2021. NCT04164901: Phase 3, multicentre, randomised, double-blind, placebo-controlled study of AG-881 in subjects with residual or recurrent grade 2 glioma with an IDH1 or IDH2 mutation. Approximately 366 participants are planned to be randomised 1:1 to receive orally administered Vorasidenib 50 mg QD or placebo. Press release published March 2023.	Agios/Celgene
AGI 5198 [179]	IDH1 R132 C, H	Reversible, allosteric competitive inhibitor to α-KG	Penetrance of BBB in mouse glioma xenografts.			Agios/Celgene
BAY 1436032 [180,181]	IDH1 R132 C, G, H, L, S	Allosteric non-competitive inhibitor	Penetrance (low): 0.06–0.38 brain to plasma ratio of BBB in a mouse model.		NCT02746081: A phase 1, open-label, non-randomised, multicentre trial of tolerance and pharmacodynamic evaluation in solid tumour with IDH1 mutation. A total of 81 patients.	Bayer
IDH 305 [182,183]	IDH1 R132 C, H	Allosteric non-competitive inhibitor	Penetrance of BBB in murine models.		NCT02381886: A phase 1, single group assignment, open-label trial for advanced malignancies harbouring IDHR132H mutations. A total of 166 patients. Regimen of 75–750 mg twice daily. Complete remission (CR) or CR with incomplete count recovery occurred in 10/37 (27%) patients with AML and 1/4 patients with myelodysplastic syndrome. Adverse events (AEs) suspected to be related to the study drug were reported in 53.7% of patients. Results published March 2023.	Novartis
AGI 6780 [184,185]	IDH2 R140Q	Allosteric non-competitive inhibitor		Unknown penetrance through the BBB.		Agios/Celgene
FT-2102 [186]	IDH1 R132 C	Competitive inhibitor		Unknown penetrance through the BBB.	NCT03684811: A phase 1b/2, non-randomised, parallel assignment, open-label study of recurrent/progressed glioma plus other tumours with IDH1 mutation. A total of 200 patients estimated.	Forma Hannover Medical School (Germany)
MFK A [184,185]	IDH1 R132 C, H	Unknown	Penetrance through BBB in mouse model shown with brain-to-plasma ratio >1.	Unknown penetrance through the BBB.		Merck
GSK 321 [187]	IDH1 R132 C, H, G	Reversible, allosteric non-competitive inhibitor		Unknown penetrance through the BBB.		GSK
ML 309 [184,185]	IDH1 R132 H	Reversible inhibitor		Unknown penetrance through the BBB.		
Ds 1001b [188]	IDH1 R132 X	Direct inhibitor	Shown to penetrate the BBB both in human and mouse xenograft models. Designed to penetrate the BBB.		NCT03030066: A phase 1, single group assignment, open-label study. A total of 47 participants. Twice daily oral administration resulted in anti-tumour activity in patients with recurrent/progressive IDH1-mutated glioma. Results published February 2023.	Daichi Sankyo

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
