# Peer review of "Mutant IDH in Gliomas: Role in Cancer and Treatment Options"

_cancers, 2023, doi:10.3390/cancers15112883_

Round 1

Reviewer 1 Report

Mutant IDH in gliomas: role in cancer and treatment options”

Brief Summary:

This is a review of the body literature that describes the current knowledge in the pathophysiology of isocitrate dehydrogenase (IDH) in glioma tumorigenesis. It also summarizes the current chemotherapeutic drugs designed to treat IUD mutant gliomas. The article divides the information into palatable bits, starting with description of the function of IDH, the oncogenic role of UDH mutation, and novel therapeutic option for IDH mutated gliomas.

Specific Comments:

P. 2 line 47 (Introduction)

  • “..., and other solid tumours.”

    • Consider adding references here.

P. 3 lines 100-101:

  • Formatting

    • Consider removing all CAPS on title

P. 3 lines 116-117:

  • “...and as such, partly explaining the slow but steady growth of IDH mutant gliomas.”

    • Is this true for all gliomas with IDH mutations? Be sure to avoid generalizations or add a reference at the end of this statement.

P. 8 lines 406-407:

  • “Currently IDH3 has not been found to be mutated in any tumours.”.

    • If this statement is true, then why add it as part of this article?.

P. 19 line 842 (Conclusion and Future Directions):

  • I would recommend adding the limitations of this article.

  • I see there are general recommendations for a call to perform research in IDH mutated gliomas. However, any specific recommendations or your future directions from your own group of researchers?

Grammar/Syntax Errors:

This article is written in English from the UK. However, there is inconsistency in certain words which are in English (US). For example:

  • Tumor and tumours both appear in the article

  • The word favors should be favours

  • Catalyzed vs catalysed.

This article is written in English from the UK. However, there is inconsistency in certain words which are in English (US). For example:

  • Tumor and tumours both appear in the article

  • The word favors should be favours

  • Catalyzed vs catalysed.

Author Response

  1. 2 line 47 (Introduction)
  • “..., and other solid tumours.”
    • Consider adding references here.

>thank you for the comment. The relevant references have been added (pag.2 line 56), and others updated.

  1. 3 lines 100-101:
  • Formatting
    • Consider removing all CAPS on title

>As suggested by the reviewer, all CAPS have been removed.

  1. 3 lines 116-117:
  • “...and as such, partly explaining the slow but steady growth of IDH mutant gliomas.”
    • Is this true for all gliomas with IDH mutations? Be sure to avoid generalizations or add a reference at the end of this statement.

>No this is not true for all IDH mutated gliomas. Thank you for pointing this out. We have removed the sentence.

  1. 8 lines 406-407:
  • “Currently IDH3 has not been found to be mutated in any tumours.”.
    • If this statement is true, then why add it as part of this article?.

>We consider this paragraph quite important, as although IDH3 has not been found mutated so far in human cancer, recent data reported in the paragraph showed that an altered expression of IDH3α plays a role in tumour initiation and progression. Moreover, our unpublished data suggest that novel and rare mutations in IDH3α might be important for tumour progression.

  1. 19 line 842 (Conclusion and Future Directions):
  • I would recommend adding the limitations of this article.
  • I see there are general recommendations for a call to perform research in IDH mutated gliomas. However, any specific recommendations or your future directions from your own group of researchers?

>We acknowledge the reviewer for the comment. We have re-written the conclusion, giving a new a more incisive perspectives from our line of research.

Grammar/Syntax Errors: 

This article is written in English from the UK. However, there is inconsistency in certain words which are in English (US). For example:

  • Tumor and tumours both appear in the article
  • The word favors should be favours
  • Catalyzed vs catalysed.

>thank you for noticing it. We have converted the entire manuscript to British English and revised the manuscript again.

Reviewer 2 Report

Mutant IDH in gliomas: role in cancer and treatment options

In this review, the authors summarize the current knowledge about the cellular and molecular consequences of IDH mutations, and the therapeutic approached developed to target IDH mutant tumours, with a focus on gliomas. Somatic mutations of the isocitrate dehydrogenase (IDH) 1 and 2 genes are highly frequent in lower-grade (WHO grades II-III) gliomas and secondary glioblastomas and were found to be an early event in gliomagenesis with profound effects on the molecular and genetic route of oncogenic progression and on clinical outcome. IDH mutations were also observed in several other cancers, including acute myeloid leukemia, T-cell lymphoma, chondrosarcoma, enchondroma, and cholangiocarcinoma. The identification of IDH mutations in multiple cancers suggests that this pathway is involved in oncogenesis.

I wish to congratulate the authors on their very interesting manuscript.  The review touches a very interesting topic in neurooncology. The most important publications from the literature was taken into account. Furthermore, the part “Novel therapeutic options for IDH mutant glioma” gives a very helpful overview concerning the given drugs in this field of research. Especially, the table with the different drugs provides a very good overview of the different given drugs used in this research.

In my opinion this manuscript is worthy of publication.

Author Response

We are really grateful to the reviewer for the positive comments.

Reviewer 3 Report

The manuscript provides a clear and comprehensive review on the field how mutant IDH alters the metabolism in glioma and the potential role of product caused by IDH mutation in cancers and the current and future treatment option. Similar review has been published but the emphasis points is different. This review cites lots of recent publications. The statement and conclusions are strongly supported by the listed citations. 

Minor concerns: 1, The authors had better draw 2-3 cartoons to describe how the IDH mutations alter the metabolic pathways. The cartoons will help the reader understand the review better.

2, In the abstract part: the therapeutic approached developed to target IDH mutant tumors. approached should be approach.

Author Response

Minor concerns: 1, The authors had better draw 2-3 cartoons to describe how the IDH mutations alter the metabolic pathways. The cartoons will help the reader understand the review better.

>We are thankful to the reviewer for the suggestion. We have added the cartoons to the manuscript.

2, In the abstract part: the therapeutic approached developed to target IDH mutant tumors. approached should be approach.

>Thank you for noticing it. We have corrected the misspelling.